# Simplified Assessment of Radioiodine Biokinetics for Thyroid Cancer Patients: A Practical Approach Using Continuous External Radiation Monitoring

**DOI:** 10.3390/diagnostics14101010

**Published:** 2024-05-14

**Authors:** Yao-Kuang Tsai, Li-Fan Lin, Cheng-Yi Cheng, Ching-Yee Oliver Wong, Wei-Hsung Wang, Daniel Hueng-Yuan Shen, Sui-Lung Su, En-Shih Chen, Tzai-Yang Chen, I-Feng Chen

**Affiliations:** 1Department of Nuclear Medicine, Tri-Service General Hospital, National Defense Medical Center, Taipei 114202, Taiwan; daylightv@gmail.com (Y.-K.T.); fanlin2@gmail.com (L.-F.L.); chengcy60@gmail.com (C.-Y.C.); chenenshih5301@yahoo.com.tw (E.-S.C.); tzaiyang@gmail.com (T.-Y.C.); 2Department of Nuclear Medicine, Taichung Armed Forces General Hospital, Taichung 411228, Taiwan; 3School of Medicine, National Defense Medical Center, Taipei 114201, Taiwan; 4Sutter Health, Sacramento, CA 95816, USA; cyowong@yahoo.com; 5Radiation Safety Office, Center for Energy Studies, Louisiana State University, Baton Rouge, LA 70803, USA; weihsung@lsu.edu; 6Department of Nuclear Medicine, Kaohsiung Veterans General Hospital, Kaohsiung 813414, Taiwan; shen8484@gmail.com; 7School of Public Health, National Defense Medical Center, Taipei 114201, Taiwan; a131419@gmail.com

**Keywords:** biokinetics, thyroid cancer, radiation exposure, radioiodine (RAI)

## Abstract

Introduction: The biokinetics of radioiodine (RAI) in thyroid cancer patients are complex. This study aims to develop a practical approach for assessing RAI biokinetics to predict patient discharge time and estimate radiation exposure to caregivers. Methods: We retrospectively reviewed data from patients with differentiated thyroid carcinoma undergoing RAI treatment. Serial radiation dose rates were dynamically collected during hospitalization and fitted to a biexponential model to assess the biokinetic features: RAI uptake fraction of thyroid tissue (***F_t_***) and effective half-life of extra-thyroid tissue (***T_et_***). Correlations with ^99m^Tc thyroid uptake ratio (TcUR), radiation retention ratio (RR), renal function, and body mass index (BMI) were analyzed. Results: Thirty-five patients were enrolled. The derived ***F_t_*** was 0.08 ± 0.06 and ***T_et_*** was 7.57 ± 1.45 h. Pearson’s correlation analysis revealed a significant association between ***F_t_*** and both TcUR and RR (*p* < 0.05), while ***T_et_*** correlated with renal function and BMI (*p* < 0.05). Conclusion: This novel and practical method assessing RAI biokinetics demonstrates consistency with other parameters and related studies, enhancing the model reliability. It shows promise in predicting an appropriate discharge time and estimating radiation exposure to caregivers, allowing for modifications to radiation protection precautions to follow ALARA principle and minimize the potential risks from radiation exposure.

## 1. Introduction

Radioiodine (RAI) has played a critical role in the treatment of differentiated thyroid cancer (DTC) for many decades, following total or near-total thyroidectomy. This approach has significantly benefited patients with high-risk cancer, leading to lowered recurrence rates and improved survival [1,2]. Nonetheless, the use of RAI presents certain challenges. One primary concern is the potential of radiation exposure to cohabitants or caregivers. Currently, there is no direct statistical evidence of harm from radiation exposure to individuals in contact with patients who have undergone iodine-131 (^131^I) treatment. However, according to the linear no-threshold (LNT) model, cancer risks from low-dose radiation exposure increase proportionally with radiation dose, without a minimum threshold [3,4,5,6,7]. Therefore, it is imperative to follow the principle of keeping radiation exposure “as low as reasonably achievable” (ALARA) in clinical practice to minimize the potential risks of radiation exposure [8]. Consequently, patients receiving RAI are typically isolated during treatment to minimize radiation exposure to others until specific release criteria are met, such as reducing retained activity to a specified level [9].

Numerous countries adopt regulations based on dose limits and constraints recommended by organizations such as the International Commission on Radiological Protection (ICRP) and the International Atomic Energy Agency (IAEA) [9,10]. For instance, the Nuclear Regulatory Commission (NRC) in the United States recommends using an external radiation dose rate as a release criterion. A patient may be discharged once the external radiation dose rate at a distance of 1 m falls below 70 microsieverts per hour (70 µSv h^−1^) [11]. Despite these regulatory frameworks, it remains challenging to accurately predict the time when a patient’s radiation dose rate meets the discharge criterion or the actual radiation exposure to individuals in the vicinity. This complexity arises from the intricate biokinetics of RAI in patients who have undergone total thyroidectomy, influenced by numerous physiological factors [12,13]. Historically, methods for studying accurate RAI biokinetics, such as those introduced by Benua or Maxon [14,15], involved complex procedures, including repeated blood and urine sample collection and whole-body external counting. Nonetheless, the practical limitations of these methods for patients undergoing RAI therapy highlight the necessity for developing a non-invasive and simplified approach to enhance dosimetric measurement. In this study, we aimed to develop a simplified and practical method to assess the RAI biokinetics in patients undergoing RAI treatment, using a continuous external radiation monitoring approach.

## 2. Materials and Methods

### 2.1. Patients

The inclusion criteria of this retrospective observational study extended to consecutively enrolled patients diagnosed with differentiated thyroid carcinoma who subsequently underwent total thyroidectomy and received their first course of RAI therapy at our hospital between 1 June 2014, and 28 February 2015. This study was exempt from IRB approval as it involved anonymous retrospective analysis of human data. Patients under the age of twenty, or those with distant metastases, recurrent disease, or insufficient information were excluded.

### 2.2. External Radiation Dose Rate Measurements and Whole-Body Radiation Retention Ratio

The external radiation dose rates of patients were detected with a built-in Geiger–Muller counter positioned on the ceiling two meters above the bed, with correction for background radiation level [12]. The distances of radiation dose rates were all corrected from 2 m into 1 m according to the IAEA’s suggestion [9]. According to the IAEA, if the distance between two people is less than 3 m, the inverse square law is not reliable. The ratio relationship for a distance less than 3 m is approximated as follows:$$D=D_{1m}x^{-1.5}$$
where *D*_1*m*_ is the radiation dose rate at 1 m, and *D* represents the radiation dose rate at a point with *x* meters away from the radiation source.

The radiation dose rates were automatically recorded at five-minute intervals throughout the patients’ hospitalization, resulting in approximately four hundred consecutive time points of recorded radiation dose rates for each patient. From these recorded dose rate data points, the whole-body radiation retention ratios (RRs) were calculated. These consecutive RR values were then utilized in the subsequent fitting process to the biexponential model. RR was defined as the ratio of the external radiation dose rate at a given time to the initial external radiation dose rate after oral ingestion of sodium iodide [^131^I], both measured at the same distance. It was calculated using the following formula:$$Whole\;body\;radiation\;retention\;ratio\;\left(RR,\;\%\right)=\left(\frac{\;\left(dose\;rate\;at\;a\;given\;time\right)-\left(background\;dose\;rate\right)}{\left(initial\;dose\;rate\;after\;RAI\;ingestion\right)-\left(background\;dose\;rate\right)}\right)\times 100$$

### 2.3. Biokinetic Modelling of RAI

The consecutive RR data were then fitted to a biexponential model for biokinetic assessment of RAI, as outlined by Zanzonico et al. [16] and the National Council on Radiation Protection and Measurements Report (NCRP) No. 155 [17]. The model is described with the following formula:$$RR=RR\;(0)\times \left[F_{t}\times {\left(\frac{1}{2}\right)}^{\frac{t}{T_{t}}}+\left(1-F_{t}\right)\times {\left(\frac{1}{2}\right)}^{\frac{t}{T_{et}}}\right]$$
where *RR* is the radiation retention ratio at 1 m from the patient at time *t* (hours) after RAI administration, with *RR(0)* representing the initial radiation retention ratio (at time 0, equal to 1), ***F_t_*** represents the RAI uptake fraction of the thyroid tissue, (1 − ***F_t_***) indicates the RAI uptake fraction of the extra-thyroid tissue, ***T_et_*** represents the effective half-life of RAI in extra-thyroid tissue, and ***T_t_*** is the effective half-life of RAI of thyroid tissue, which is fixed at a constant of 175.2 h as described by the NRC [11].

Approximately four hundred consecutive time points of detected radiation dose rates were recorded, yielding corresponding RRs. These sequential ratios were then fitted to a biexponential model. We utilized Microsoft^®^ Excel Generalized Reduced Gradient (GRG) Nonlinear Solver to minimize the sum of squared differences between each recorded RR and the corresponding function value of the equation, thereby obtaining the most accurate functional curve for each patient’s metabolism of RAI (Figure 1a). However, fluctuations in recorded RR data may arise due to unrestricted patient mobility within the isolation ward. To address this, we applied Tukey’s method to identify and eliminate outliers [18]. This method partitions the data into quartiles, where Q_1_ comprises values equal to or greater than 1/4 of the data, Q_2_ includes values equal to or greater than 1/2 of the data, and Q_3_ includes values equal to or greater than 3/4 of the data. The interquartile range (IQR) is calculated as the difference between Q_3_ and Q_1_. According to Tukey’s rule, outliers are defined as values that deviate by more than 1.5 times the IQR from the quartiles, either below (Q_1_ − 1.5 × IQR) or above (Q_3_ + 1.5 × IQR). After removing the outliers, the data were refitted to the biexponential equation, resulting in a more realistic functional curve for estimating the patients’ RAI biokinetics. With ***T_et_*** (the effective half-life of RAI in extra-thyroid tissue) and ***F_t_*** (the RAI uptake fraction of the thyroid tissue) values as variables, we utilized GRG Nonlinear Solver to derive these biokinetic parameters for each patient (Figure 1b).

### 2.4. Radiation Exposure Estimation

Upon discharge, we promptly collected patients’ radiation data and calculated the biokinetics (***F_t_*** and ***T_et_***). Subsequently, the delineated biexponential RAI biokinetic curves were extrapolated from the time of release to infinity, with the area under curve (AUC) representing the estimation of maximal radiation exposure for an individual consistently positioned at a one-meter distance from the patient (Figure 1c).
$${Total\;exposure\;radiation\;dose\;\left(at\;1\;meter\right)\;after\;release=D}_{1m}\left(0\right)\int_{t-release}^{\infty }RR(0)\left[F_{t}\times {\left(\frac{1}{2}\right)}^{\frac{t}{T_{t}}}+\left(1-F_{t}\right)\times {\left(\frac{1}{2}\right)}^{\frac{t}{T_{et}}}\right]dt$$
where *D*_1*m*_(0) is the radiation dose rate at 1 m at initial (*t* = 0) and *t-release* represents the time (hours) the patient spent in the isolation ward before release.

### 2.5. 99^m^Tc-pertechnetate Uptake Ratio

Thyroid scans using technetium-99m pertechnetate (^99m^Tc) were conducted on all patients prior to RAI treatment to assess the abundance of thyroid remnants and calculate the ^99m^Tc uptake ratio (TcUR) (Figure 2). This ratio was subsequently utilized to validate the derived biokinetic parameter ***F_t_*** (the RAI uptake fraction of the thyroid tissue). The TcUR was calculated using the formula:Tc  99muptake ratio TcUR, %=Thyroid remnants counts−background counts differences in counts of the syringe before and after injection×100

### 2.6. Estimated Glomerular Filtration Rate

Estimated glomerular filtration rate (eGFR) was calculated using the modification of diet in renal disease (MDRD) equation. An eGFR of >90 mL/min/1.73 m^2^ was considered normal kidney function.

### 2.7. Statistical Analysis

All statistical analyses were conducted using SPSS, version 20.0 (SPSS Inc., Chicago, IL, USA). Continuous variables are presented as mean ± standard deviation (SD). Correlations between continuous variables were assessed using Pearson’s correlation coefficients. The analysis included the examination of ***F_t_*** in relation to TcUR and whole-body radiation retention ratio at release time (RR_release_). Additionally, ***T_et_*** was analyzed in relation to age, sex, BMI, serum creatinine level, blood urea nitrogen (BUN) level, and eGFR. A probability value of *p* < 0.05 was considered statistically significant.

## 3. Results

A total of 95 patients diagnosed with differentiated thyroid cancer, who had undergone total thyroidectomy and subsequently received RAI therapy at our hospital between 1 June 2014 and 28 February 2015, were consecutively enrolled in this study. Among these patients, 37 were excluded due to recurrent disease, 1 due to distant metastases, and 22 due to insufficient information. The final analysis was conducted on the data of 35 patients, among whom 33 had been diagnosed with papillary thyroid carcinoma (PTC), 1 with follicular thyroid carcinoma (FTC), and 1 with Hurthle cell carcinoma. The mean age of the study participants was 46.91 ± 14.85 years (range 26–76 years). The mean administered dose of RAI was 3.74 ± 1.17 GBq (range 1.11–5.55 GBq). The characteristics of our study population are presented in Table 1.

The mean derived RAI uptake fraction of thyroid tissue (***F_t_***) was 0.08 ± 0.06 (range 0.006–0.288) (Table 1). Pearson’s correlation analysis revealed a significant positive correlation between ***F_t_*** and both TcUR (r = 0.769, *p* < 0.001) and RR_release_ (r = 0.750, *p* < 0.001) (Table 2 and Figure 3). The mean derived effective half-life of RAI in extra-thyroid tissues (***T_et_***) was 7.57 ± 1.45 h (range 4.180–10.228 h) (Table 1). A significant negative correlation was observed between ***T_et_*** and eGFR (r = −0.400, *p* = 0.017), as well as a significant positive correlation between ***T_et_*** and creatinine level (r = 0.629, *p* < 0.001), BUN level (r = 0.371, *p* = 0.024) and BMI (r = 0.374, *p* = 0.022) (Table 3).

## 4. Discussion

In this study, we introduce a novel and simplified method for investigating the biokinetic parameters of RAI in thyroid cancer patients. This method holds potential applications in clinical practice to conveniently predict patients’ release time and assess the radiation exposure to caregivers after release from the isolation ward. Being noninvasive and practical, the method allows patients to move freely within the ward, eliminating the need for additional work or complex procedures. According to the review of previous studies, the assessment of RAI biokinetics has posed consistent challenges. Traditionally, these assessment methods were often intricate and occasionally invasive. For example, Benua et al. [14,19] employed a monitoring approach involving sequential collection of blood samples (at 2, 4, 24, 48, 72, and 96 h), urine samples (at 24, 48, 72, and 96 h), and whole-body counting (at 0, 2, 4, 24, 48, 72, and 96 h) post-RAI administration. While this method can accurately determine the maximum safe RAI dosage for bone marrow protection, its complexity often renders it impractical for clinical use. In contrast, our study presents a method that offers clinical applicability for estimating RAI biokinetics without additional invasive procedures. Our approach yielded a comprehensive dataset for each patient, comprising approximately 400 continuous points of radiation dose rate data. This voluminous dataset allowed precise curve fitting and derivation of biokinetic parameters, potentially resulting in a comparably accurate estimation of true RAI biokinetics in patients. This method holds promise for practical application in every clinical patient.

After total thyroidectomy, residual thyroid tissue is typically minimal. In a study by Tenhunen et al. [20], a biexponential model was employed to estimate the thyroidal uptake, reporting a minimal value of 0.01 (range 0.000–0.051). Various studies have also used different small values to define the thyroid uptake fraction of RAI. The NRC Regulatory Guide 8.39 proposed a value of 0.05, while the American Thyroid Association (ATA) in 2011 and another study by Liu et al. [21] suggested 0.02 [11,21,22]. In our current study, the derived thyroid uptake fraction of RAI (***F_t_***) was similarly small (0.08 ± 0.06, range 0.006–0.288). However, despite the small value, the thyroid uptake fraction should be considered cautiously and calculated separately due to the significantly longer half-life of iodine in thyroid tissue compared with other organs. Our study, utilizing the biexponential model, clearly illustrated a strong positive correlation between ***F_t_*** and the RR_release_ (r = 0.750, *p* < 0.001), underscoring the importance of RAI uptake in thyroid remnants in influencing RAI clearance and patients’ release time. This hypothesis is further supported by a previous study by Tenhunen et al. [20], which demonstrated the preferred model for evaluating RAI kinetics in thyroid carcinoma patients to be biexponential with a two-component design to consider iodine trapped within the thyroid remnants and in extra-thyroid tissues.

Moreover, to validate the accuracy of thyroidal uptake fraction (***F_t_***) derived from the biexponential model, we assessed the correlation between ***F_t_*** and TcUR. TcUR, calculated from the ^99m^Tc thyroid scan, has been shown to accurately evaluate the abundance of thyroid remnants in post-operative thyroid cancer patients [23,24,25]. Our findings indicated a strong positive correlation between ***F_t_*** and TcUR (r = 0.769, *p* < 0.001), further affirming the reliability of our biexponential RAI biokinetics model.

Renal clearance predominantly influences the excretion of RAI, as iodine is primarily excreted in urine [26]. Previous research has also demonstrated a correlation between the effective whole-body half-life of RAI and factors such as eGFR, body height, and weight [27]. In our investigation, the effective half-life of RAI in extra-thyroid tissue (***T_et_***) was found to be independently associated with patients’ eGFR and BMI, consistent with prior findings. Remarkably, ***T_et_*** in our study population derived using the biexponential model was calculated to be 7.57 ± 1.45 h (range 4.180–10.228 h), closely aligning with the value for the same parameter stated in the NRC Regulatory Guide 8.39 (7.69 h) [11]. This consistency further strengthens the credibility of our model.

Radiation exposure to cohabitants and caregivers of released patients remains a significant concern. Our approach establishes a simplified and practical model for estimating patients’ biokinetics (***F_t_*** and ***T_et_***) of RAI, which further facilitates predicting an appropriate discharge time and assessing radiation exposure to caregivers post-release. For example, in Figure 1, the case involves an estimated maximal radiation exposure post-release of 2.9 mSv, complying with the ICRP recommended dose constraint of 5 mSv per therapy episode for relatives and caregivers [10]. On the other hand, if the estimated radiation exposure exceeds the limit, the information can guide individualized adjustments of personalized radiation protection precautions to align with the goal of ALARA, resulting in the minimization of potential risks from radiation exposure. This method holds potential for widespread application in every clinical patient.

The present study was subject to several limitations. Firstly, we did not use the traditional method of collecting blood and urine samples to evaluate RAI biokinetics, which may have lowered the accuracy of the results. Alternatively, however, our study collected a voluminous dataset of serial radiation dose rates, potentially compensating for this limitation. Moreover, previous research has shown that the effective half-life of RAI is associated with eGFR, and the eGFR data in our study were estimated based on serum creatinine levels, which may not fully reflect real renal clearance. Nevertheless, considering that all our patients had serum creatinine levels within the normal range, this may not have significantly affected the data integrity. Finally, the retrospective observational design and relatively small sample size restrict the generalizability of these results. Further prospective research may be warranted.

## 5. Conclusions

Our study introduces a novel and practical method for assessing RAI biokinetics (***F_t_*** and ***T_et_***). The consistency of calculated values with other parameters and related studies enhances the reliability of our model. Moreover, this model shows promise in predicting an appropriate discharge time and estimating radiation exposure to caregivers post-release, allowing for modifications to radiation protection precautions to adhere to the ALARA principle, resulting in the minimization of potential risks from radiation exposure. Our method holds potential for widespread application in every clinical patient.

## Figures and Tables

**Figure 1 diagnostics-14-01010-f001:**
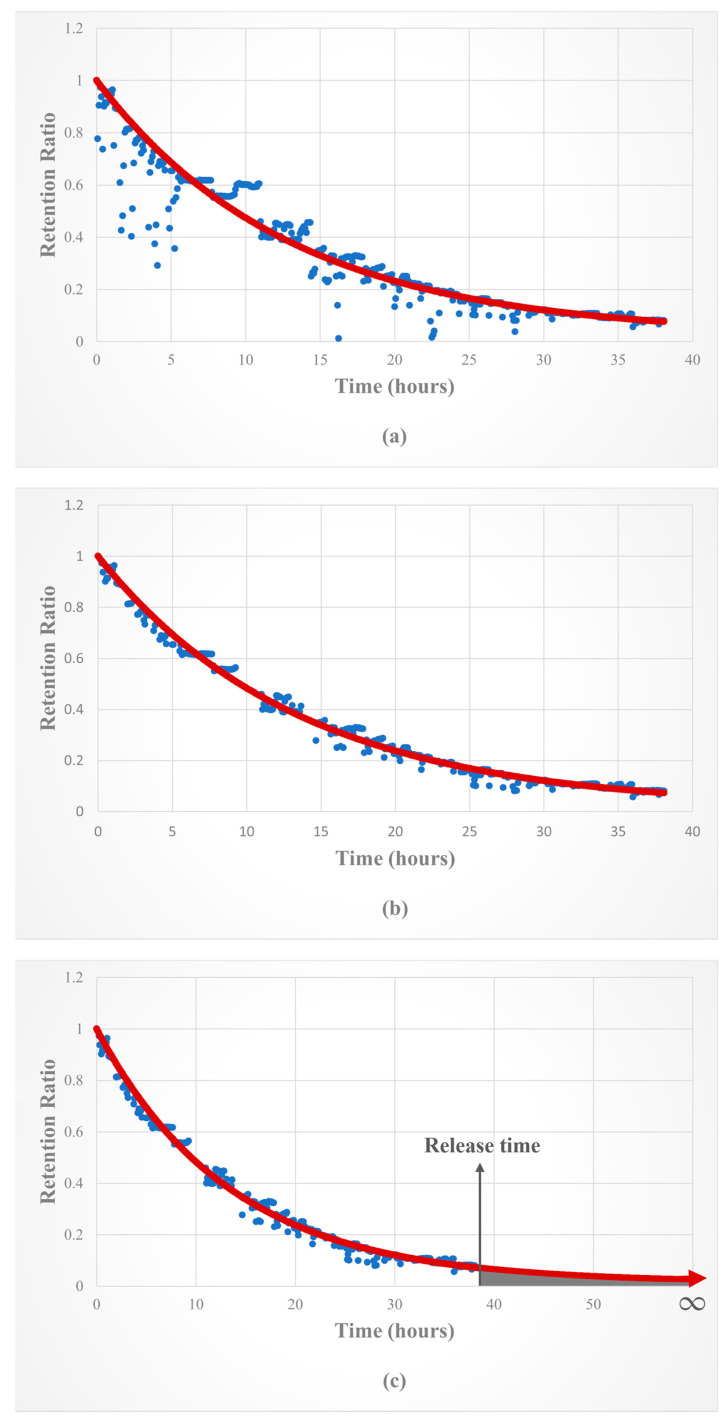
Time-radiation retention ratio curve and biexponential fitting for radiation exposure estimation. A 67-year-old female diagnosed with differentiated thyroid cancer (DTC) after undergoing total thyroidectomy was admitted for radioiodine (RAI) treatment with a dose of 4.44 GBq. Radiation dose rates were measured every five minutes and converted to radiation retention ratios (RRs, blue points). These data were subsequently fitted to a biexponential functional curve (red curve). Some fluctuations in radiation ratios were observed at certain points due to the patient’s unrestricted mobility within the isolation ward (**a**). Outlier data were identified and excluded using Tukey’s method. The RAI uptake fraction of the thyroid tissue (***F_t_***) and the effective half-life of extra-thyroid tissue (***T_et_***) were then determined. In this case, ***F_t_*** was 0.017 and ***T_et_*** was 9.318 h (**b**). After release (approximately 38 h post-treatment), the biexponential curve was extrapolated to infinity. The area under the curve (AUC) represents the maximal radiation exposure for a person positioned consistently one meter away from the patient post-release. In this case, the approximation is calculated to be 2.9 mSv (**c**).

**Figure 2 diagnostics-14-01010-f002:**
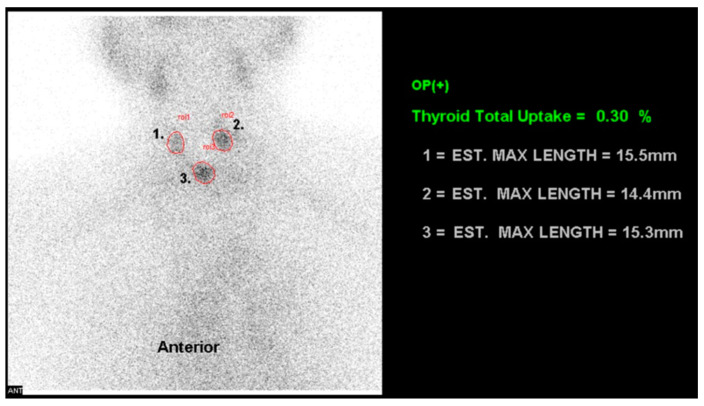
^99m^Tc-pertechnetate thyroid scan and ^99m^Tc uptake ratio. A thyroid scan using ^99m^Tc-pertechnetate (^99m^Tc) was conducted several days prior to radioiodine treatment in a 73-year-old male. Regions of interest (ROIs) were delineated to assess the residual thyroid tissue and calculate the ^99m^Tc uptake ratio (TcUR). In this case, the TcUR was 0.3%, consistent with the patient’s history of post-total thyroidectomy.

**Figure 3 diagnostics-14-01010-f003:**
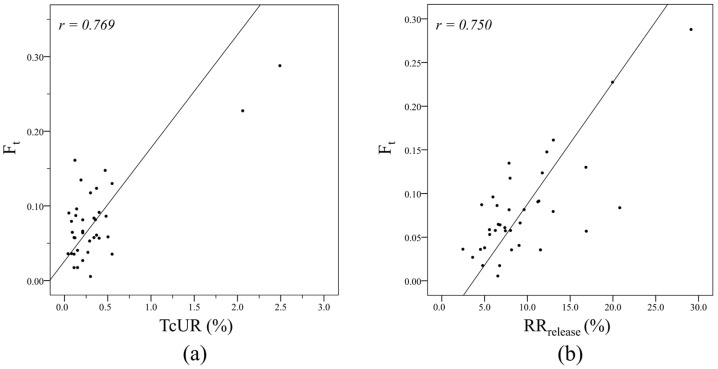
Relationship between the RAI uptake fraction of the thyroid tissue (***F_t_***) and ^99m^Tc uptake ratio (TcUR) (**a**) and whole-body retention ratio at release time (RR_release_) (**b**). Pearson’s correlation analysis revealed a strong positive correlation between ***F_t_*** and both TcUR (r = 0.769, *p* < 0.001) (**a**) and RR_release_ (r = 0.750, *p* < 0.001) (**b**).

**Table 1 diagnostics-14-01010-t001:** Characteristics of enrolled thyroid cancer patients.

Characteristic	N (%)	Mean ± SD	Range
**Age**		46.91 ± 14.85	26–76
**Sex**			
Women	23 (65.7%)		
Men	12 (34.3%)		
**Pathology**			
Papillary carcinoma	33 (94.3%)		
Follicular carcinoma	1 (2.9%)		
Hurthle cell carcinoma	1 (2.9%)		
**T stage**			
1	11 (31.4%)		
2	12 (34.3%)		
3	10 (28.6%)		
4	2 (5.7%)		
**N stage**			
No lymph node metastasis (N0)	19 (54.3%)		
Lymph node metastases (N1)	16 (45.7%)		
**Preparation of TSH stimulation**			
rh-TSH ^a^ administration	30 (85.7%)		
T4 withdrawal	5 (14.3%)		
**Administered dose of RAI ^b^ (GBq)**	35	3.74 ± 1.17	1.11–5.55
1.11 GBq	5 (14.3%)		
3.7 GBq	12 (34.3%)		
4.44 GBq	17 (48.6%)		
5.55 GBq	1 (2.9%)		
**RR_release_ ^c^ (%)**		9.61 ± 5.57	2.50–29.12
**TcUR ^d^ (%)**		0.36 ± 0.50	0.04–2.49
**BMI**		23.23 ± 3.39	17.69–33.41
**TSH (mIU/mL)**		164.01 ± 80.80	38.61–375.00
**Creatinine (mg/dL)**		0.82 ± 0.17	0.5–1.2
**BUN ^e^ (mg/dL)**		10.37 ± 2.28	7–15
**eGFR ^f^ (mL/min per 1.73 m^2^)**		87.62 ± 17.51	61.3–129.1
**Derived biokinetic parameters of RAI**			
***F_t_*** (RAI uptake fraction of the thyroid tissue)		0.08 ± 0.06	0.006–0.288
***T_et_*** (effective half-life of RAI in extra-thyroid tissue)		7.57 ± 1.45	4.180–10.228

^a^: rh-TSH: recombinant human thyroid-stimulating hormone; ^b^: RAI: radioiodine (iodine-131); ^c^: RR_release_: whole-body radiation retention at release time; ^d^: TcUR: ^99m^Tc uptake ratio; ^e^: BUN: blood urea nitrogen; ^f^: eGFR: estimated glomerular filtration rate.

**Table 2 diagnostics-14-01010-t002:** Coefficients of correlation between ***F_t_*** ^a^ and TcUR and RR_release_.

Variable	Correlation Coefficient, r	*p* Value
TcUR ^b^	0.769	<0.001
RR_release_ ^c^	0.750	<0.001

^a^: ***F_t_***: RAI uptake fraction of the thyroid tissue; ^b^: TcUR: ^99m^Tc uptake ratio; ^c^: RR_release_: whole-body retention ratio at release time.

**Table 3 diagnostics-14-01010-t003:** Coefficients of correlation between ***T_et_*** ^a^ and BMI, age, BUN, creatinine, and eGFR.

Variable	Correlation Coefficient, r	*p* Value
Age	0.160	0.345
BMI ^b^	0.374	0.022 *
BUN ^c^ level	0.371	0.024 *
Creatinine level	0.629	<0.001 ***
eGFR ^d^	−0.400	0.017 *

*: *p* < 0.05; ***: *p* < 0.001; ^a^: ***T_et_***: effective half-life of RAI in extra-thyroid tissue; ^b^: BMI: body mass index; ^c^: BUN: blood urea nitrogen; ^d^: eGFR: estimated glomerular filtration rate.

## Data Availability

The data presented in this study are available on request from the corresponding author. The data are not publicly available due to privacy regulations.

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
