# Peer review of "Simplified Assessment of Radioiodine Biokinetics for Thyroid Cancer Patients: A Practical Approach Using Continuous External Radiation Monitoring"

_diagnostics, 2024, doi:10.3390/diagnostics14101010_

Round 1

Reviewer 1 Report

Comments and Suggestions for Authors

The authors described a novel approach to measure the radiation of patients after receiving RAI, which could cause hazard to other nearby people and caregivers.  

1.       My first observation is that I don’t see any reference on the fact that RAI patients may jeopardise their surroundings

2.       Figure 1 is an animation and is obsolete, as the process is already explained in the paper

3.       In Figure 4a the regression line doesn’t seem convincing. It seems that most samples are grouped in the lower left angle, while the other two samples seem to be outliers

4.       Is it necessary to collect 400 measurings per patient? Seems to be a bit excessive. Over which period of time did you take these measurements?

Author Response

Dear reviewer:

I am very grateful to your comments for the manuscript. According with your advice, we amended the relevant part in manuscript. Some of your questions were answered below.

1. My first observation is that I don’t see any reference on the fact that RAI patients may jeopardise their surroundings.

Author response:

    We greatly appreciate your comments. We are fully aware of the importance of minimizing the risks associated with radiation exposure, including the stochastic effects, which may lead to radiation-induced cancer, genetic mutations, and teratogenic effects. Therefore, it is crucial to cautiously estimate even low-dose radiation exposure. Our study aims to introduce a practical method for estimating radiation exposure to ensure that caregivers and cohabitants adhere to the ALARA (As Low As Reasonably Achievable) principle.

    In response to your inquiry, we have meticulously reviewed the existing literature and have found no direct statistical evidence of harm resulting from exposure to individuals in contact with patients who have undergone RAI treatment. However, several studies (references 1-5, now referenced as 3-7 in the revised manuscript) suggest that the Linear No-Threshold (LNT) model indicates an increased risk of cancer from low-dose radiation exposure, proportionally with the radiation dose, without a minimum threshold [1-5]. Consequently, we have revised the introduction (lines 39-48) and included these references in the revised manuscript.

2. Figure 1 is an animation and is obsolete, as the process is already explained in the paper.

Author response:

    We are grateful for your professional advice, and we agree with your suggestion. As a result, we have removed Figure 1 to enhance the clarity of the manuscript.

3. In Figure 4a the regression line doesn’t seem convincing. It seems that most samples are grouped in the lower left angle, while the other two samples seem to be outliers.

Author response:

    Thank you for providing these important comments. The unsatisfactory distribution observed in Figure 4a (marked as 3a in the revised manuscript) may be attributed to our relatively small sample size and the insufficient diversity within the patient group. Among the enrolled patients, two individuals were diagnosed with advanced thyroid cancer, resulting in larger remnants after surgery. This led to higher TcUR values and corresponding elevated Ft levels. Conversely, the remaining patients exhibited minimal remnants and demonstrated lower TcUR values, placing them in the lower left region. The absence of patients with intermediate TcUR values created a gap in the middle region of the scatter plot, giving it a less than optimal appearance. While this limitation may not significantly impact the results, we have nonetheless acknowledged it in the section discussing limitations (lines 299-300).

4. Is it necessary to collect 400 measurings per patient? Seems to be a bit excessive. Over which period of time did you take these measurements?

Author response:

    Thank you for this valuable assessment. The Geiger-Muller ceiling counter automatically detected the radiation dose rates every 5 minutes during the patients’ hospitalization (about 38 hours). The 400 measurings were automatically generated with no additional work. We believe these continuous measurings can reflect patients’ radioiodine metabolism over time. To enhance the automatic advantage of our model, we have revised the paragraph of methods (line 91).

* . References in author response to question 1 (reference 3-7 in the revised manuscript)

  1. Council, N.R. Health Risks from Exposure to Low Levels of Ionizing Radiation: BEIR VII Phase 2; The National Academies Press: Washington, DC, 2006; p. 422.
  2. Brenner, D.J.; Doll, R.; Goodhead, D.T.; Hall, E.J.; Land, C.E.; Little, J.B.; Lubin, J.H.; Preston, D.L.; Preston, R.J.; Puskin, J.S.; et al. Cancer risks attributable to low doses of ionizing radiation: assessing what we really know. Proceedings of the National Academy of Sciences of the United States of America 2003, 100, 13761-13766, doi:10.1073/pnas.2235592100.
  3. No, N.C. Implications of recent epidemiologic studies for the linear nonthreshold model and radiation protection; NCRP: 2018.
  4. Shore, R.E.; Beck, H.L.; Boice, J.D.; Caffrey, E.A.; Davis, S.; Grogan, H.A.; Mettler, F.A.; Preston, R.J.; Till, J.E.; Wakeford, R.; et al. Implications of recent epidemiologic studies for the linear nonthreshold model and radiation protection. J Radiol Prot 2018, 38, 1217-1233, doi:10.1088/1361-6498/aad348.
  5. Agency, U.E.P. EPA radiogenic cancer risk models and projections for the US population. 2011.

Reviewer 2 Report

Comments and Suggestions for Authors

The authors introduced a new method for investigating the biokinetic parameters of iodine in thyroid cancer patients treated with Radioiodine. It seems that this manuscript is well-designed and written and provides good content for the readers of this journal. I believe that reading/citing these articles may help improve the quality of this study (PMIDs: 27385884, 27555655).

Author Response

    The authors introduced a new method for investigating the biokinetic parameters of iodine in thyroid cancer patients treated with Radioiodine. It seems that this manuscript is well-designed and written and provides good content for the readers of this journal. I believe that reading/citing these articles may help improve the quality of this study (PMIDs: 27385884, 27555655).

Author response:

    We express our sincere gratitude for your acknowledgment and for the articles you recommended. We have carefully reviewed these two exemplary studies, which have served as valuable inspiration and guidance for the methodology of our ongoing research.

    The first research examined the impact of exercise on reducing radiation exposure in differentiated thyroid cancer patients receiving radioiodine therapy post-thyroidectomy. Results showed exercise significantly reduced radiation levels, suggesting sweating aids in radioiodine excretion. This finding could shorten quarantine periods, especially in patients unable to tolerate excessive hydration due to health issues.

    The other research used TLD dosimetry to measure radiation doses during OPG imaging. Results highlighted variations in doses across head and neck organs. It can be concluded that we can put more TLD chips inside each organ, when we want to estimate the effective dose accurately.

    While we acknowledge the merit of these studies, we have determined that their inclusion may not align with the focus of our current manuscript. Nevertheless, we wish to convey our deep appreciation for the insights gained from these well-designed studies. Your recommendation has proven immensely beneficial to our work, and we are truly thankful for your support and encouragement.

Reviewer 3 Report

Comments and Suggestions for Authors

This is a retrospective study on patients with differentiated thyroid cancer being treated with radioactive iodine, aiming to develop a novel approach for assessing RAI biokinetics to estimate radiation exposure to others and predict optimal patient discharge time.

The results are clearly presented. However, in order for the readers to fully appreciate the progress potentially made by the future use of this newly proposed method it is essential to describe in a lot more detail what is the method used now for this purpose. Otherwise it is impossible for a non-nuclear medicine doctor to assess how practical would it be to use this approach.

Author Response

    This is a retrospective study on patients with differentiated thyroid cancer being treated with radioactive iodine, aiming to develop a novel approach for assessing RAI biokinetics to estimate radiation exposure to others and predict optimal patient discharge time.

    The results are clearly presented. However, in order for the readers to fully appreciate the progress potentially made by the future use of this newly proposed method it is essential to describe in a lot more detail what is the method used now for this purpose. Otherwise it is impossible for a non-nuclear medicine doctor to assess how practical would it be to use this approach.

Author response:

    We greatly appreciate your professional advice. It is difficult to assess a patient’s biokinetics of RAI. Currently, if we want to assess a patient’s biokinetics of RAI, the assessment methods may be intricate and occasionally invasive, such as sequential collection of blood samples, urine samples, and continuous whole-body counting post-RAI administration, which was challenging in every clinical patient. Therefore, we introduced a novel and simplified method for studying RAI biokinetics.

    Using this simplified and practical method, we can assess patients biokinetic features and further estimate the total exposure to cohabitants and caregivers after the patients' release. Based on the estimated values, we can guide individualized adjustments of personalized radiation protection precautions to align with the ALARA principle and ensure effective radiation protection and minimize the potential risk of radiation exposure to the cohabitants and caregivers of RAI patients.

    We are grateful for your invaluable suggestions. It is crucial to emphasize the practical and clinical utility of our method, enabling non-nuclear medicine doctors to easily assess it. Accordingly, we have revised the sections of discussion (lines 247 and 288-290) and conclusion (lines 308-309) in accordance with your suggestion.